# Sparsity in Continuous-Depth Neural Networks

**Hananeh Aliee**
Helmholtz Munich

**Till Richter**
Helmholtz Munich

**Mikhail Solonin**[*]
Technical University of Munich

**Ignacio Ibarra**
Helmholtz Munich

**Fabian Theis**
Technical University of Munich
Helmholtz Munich

**Niki Kilbertus**
Technical University of Munich
Helmholtz AI, Munich

{hananeh.aliee,till.richter,ignacio.ibarra,fabian.theis,niki.kilbertus}
@helmholtz-muenchen.de

## Abstract

Neural Ordinary Differential Equations (NODEs) have proven successful in learning dynamical systems in terms of accurately recovering the observed trajectories. While different types of sparsity have been proposed to improve robustness, the generalization properties of NODEs for dynamical systems beyond the observed data are underexplored. We systematically study the influence of weight and feature sparsity on forecasting as well as on identifying the underlying dynamical laws. Besides assessing existing methods, we propose a regularization technique to sparsify "input-output connections" and extract relevant features during training. Moreover, we curate real-world datasets consisting of human motion capture and human hematopoiesis single-cell RNA-seq data to realistically analyze different levels of out-of-distribution (OOD) generalization in forecasting and dynamics identification respectively. Our extensive empirical evaluation on these challenging benchmarks suggests that weight sparsity improves generalization in the presence of noise or irregular sampling. However, it does not prevent learning spurious feature dependencies in the inferred dynamics, rendering them impractical for predictions under interventions, or for inferring the true underlying dynamics. Instead, feature sparsity can indeed help with recovering sparse ground-truth dynamics compared to unregularized NODEs.

## 1 Introduction

Extreme over-parameterization has shown to be part of the success story of deep neural networks [52, 5, 4]. This has been linked to evidence that over-parameterized models are easier to train, perhaps because they develop "convexity-like" properties that help convergence of gradient descent [10, 14]. However, over-parameterization comes at the cost of additional computational footprint during training and inference [13]. Therefore, motivation for sparse neural networks are manifold and include: (i) imitation of human learning, where neuron activity is typically sparse [2], (ii) computational efficiency (speed up, memory reduction, scalability) [20], (iii) interpretability [51], as well as (iv) avoiding overfitting or improving robustness [6].

Enforcing sparsity in model weights, also called model pruning, and its effects for standard predictive modelling tasks have been extensively studied in the literature [20]. However, sparsity in continuous-depth neural nets for modeling dynamical systems and its effects on generalization properties is still

---

[*]Work done while at TUM. MS is currently employed by J.P. Morgan Chase & Co.; mikhail.solonin@jpmorgan.com

36th Conference on Neural Information Processing Systems (NeurIPS 2022).

underexplored. Neural Ordinary Differential Equations (NODEs) [11] have been introduced as the limit of taking the number of layers in residual neural networks to infinity, resulting in continuous-depth (or continuous-time) neural nets. NODEs were originally predominantly used for predictive tasks such as classification or regression, where they learn *some* dynamics serving the predictive task. These networks have also shown great promise for modeling noisy and irregular time-series. More recently, it has been stated that NODEs can also be thought of as learning *the underlying* dynamics (i.e., the ODE) that actually govern(s) the evolution of the observed trajectory. In the prior scenario (predictive performance), a large body of literature has focused on regularizing the number of model evaluations required for a single step of the ODE solver for improving the efficiency [16, 22, 17, 39, 18, 38]. These regularization techniques rely on learning one out of many possible equivalent dynamical systems giving rise to the same ultimate performance, but are easier and faster to solve. Crucially, these regularizers do not explicitly target sparsity in the model weights or in the features used by the model.

In the latter scenario (inferring dynamical laws), we assume the existence of a ground truth ODE that governs the dynamics of the observed trajectories. Here, two types of sparsity may be of interest. First, motivated by standard network pruning, sparsity in model weights (*weight sparsity* or *model sparsity*) can reduce computational requirements for inference [28], see Figure 1(top). Second, it has been argued that *feature sparsity*, i.e., reducing the number of inputs a given output of the NODE relies on, can improve identifiability of the underlying dynamical law [3, 7, 30], see Figure 1(bottom). The motivation for feature sparsity often relies on an argument from causality, where it is common to assume sparse and modular causal dependencies between variables [45, 54].

In this paper, we empirically assess the impact of various types of sparsity-enforcing methods on the performance of NODEs on different types of OOD generalization for both prediction and for inferring dynamical laws on real-world datasets. While most existing methods target weight sparsity, we highlight that feature sparsity can improve interpretability and dynamical system inference by proposing a new regularization technique for continuous-time models. Specifically, our main contributions are the following:

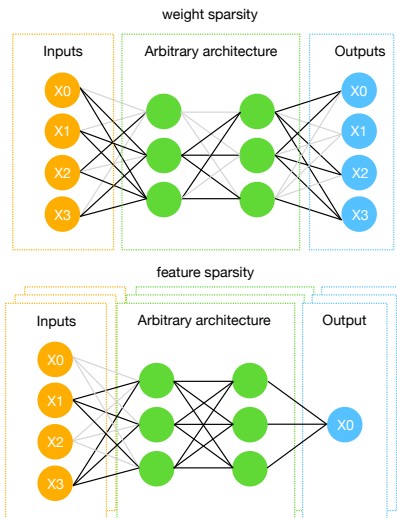

Figure 1: Model vs feature sparsity

- We propose PathReg[2], a differentiable L0-based regularizer that enforces sparsity of "input-output paths" and leads to both feature and weight sparsity.
- We extend L0 regularization [31] and LassoNet [26] to NODEs and perform extensive comparisons with existing models including vanilla NODE [11], C-NODE [3], GroupLasso for NODEs [7], and our PathReg.
- We define a useful metric for evaluating feature sparsity, demonstrate the differences between weight and feature sparsity, and explore whether and when one implies the other.
- To cover at least part of the immense diversity of time-series problems, each of which implying different types of OOD challenges, we curate large, real-world datasets consisting of human motion capture (`mocap.cs.cmu.edu`) as well as human hematopoiesis single-cell RNA-seq [32] data for our empirical evaluations.

## 2 Background

### 2.1 Continuous-depth neural nets

Among the plethora of deep learning based methods to estimate dynamical systems from data [29, 44, 27, 38], we focus on Neural Ordinary Differential Equations (NODEs). In NODEs, a neural network with parameters $\theta$ is used to learn the function $f_\theta \approx f$ from data, where $f$ defines a (first

---

[2]The python implementation is available at: https://github.com/theislab/PathReg

order) ODE in its explicit representation $\dot{X} = f(X, t)$ [11]. Starting from the initial observation $X(a)$ at some time $t = a$, an explicit iterative ODE solver is deployed to predict $X(t)$ for $t \in (a, b]$ using the current derivative estimates from $f_\theta$. The parameters $\theta$ are then updated via backpropagation on the mean squared error (MSE) between predictions and observations. As discussed extensively in the literature, NODEs can outperform traditional ODE parameter inference techniques in terms of reconstruction error, especially for non-linear dynamics $f$ [11, 15]. In particular, one advantage of NODEs over previous methods for inferring non-linear dynamics such as SINDy [9] is that no dictionary of non-linear basis functions has to be pre-specified.

A variant of NODEs for second order systems called SONODE exploits the common reduction of higher-order ODEs to first order systems [37]. The second required initial condition, the initial velocity $\dot{X}(a)$, is simply learned from $X(a)$ via another neural network in an end-to-end fashion. Our experiments build on the public NODE and SONODE implementations. However, our framework is readily applicable to most other continuous-depth neural nets including Augmented NODE [15], latent NODEs [44], and neural stochastic DEs [27, 48].

## 2.2 Sparsity and generalization

Two prominent motivations for enforcing sparsity in over-parameterized deep learning models are (a) pruning large networks for efficiency (speed, memory), (b) as a regularizer that can improve interpretability or prevent overfitting. In both cases, one aims at preserving (as much as possible) i.i.d. generalization performance, i.e., performance on unseen data from the same distribution as the training data, compared to a non-sparse model [20].

Another line of research has explored the link between generalizability of neural nets and causal learning [45], where generalization outside the i.i.d. setting, is conjectured to require an underlying *causal model*. Deducing true laws of nature purely from observational data could be considered an instance of inferring a causal model of the world. Learning the correct causal model enables accurate predictions not only on the observed data (next observation), but also under distribution shifts for example under interventions [45]. A common assumption in causal modeling is that each variable depends on (or is a function of) *few* other variables [45, 54, 36, 8]. In the ODE context, we interpret the variables (or their derivatives) that enter a specific component $f_i$ as causal parents, such that we can write $\dot{X}_i = f(pa(X_i), t)$ [35]. Thus, feature sparsity translates to each variable $X_i$ having only few parents $pa(X_i)$, which can also be interpreted as asking for "simple" dynamics. Since weight sparsity as well as regularizing the number of function evaluations in NODEs can also be considered to bias them towards "simpler dynamics", these notions are not strictly disjoint, raising the question whether one implies the other.

In terms of feature sparsity, Aliee et al. [3], Bellot et al. [7] study *system identification* and causal structure inference from time-series data using NODEs. They suggest that enforcing sparsity in the number of causal interactions improves parameter estimation as well as predictions under interventions. Let us write out $f_\theta$ as a fully connected net with $L$ hidden layers parameterized by $\theta := (W^l, b^l)_{l=1}^{L+1}$ as

$$f_\theta(X) = W^{L+1}\sigma(\ldots\sigma(W^2\sigma(W^1X + b^1) + b^2)\ldots) \tag{1}$$

with element-wise activation function $\sigma$, $l$-th layer weights $W^l$, and biases $b^l$. Aliee et al. [3] then seek to reduce the overall number of parents of all variables by attempting to cancel all contributions of a given input on a given output node through the neural net. In a linear setting, where $\sigma(x) = x$, the regularization term is defined by[3]

$$\|A\|_{1,1} = \|W^{L+1}\ldots W^1\|_{1,1} \tag{2}$$

where $A_{ij} = 0$ if and only if the $i$-th output is constant in the $j$-th input. In the non-linear setting, for certain $\sigma(x) \neq x$, the regularizer $\|A\|_{1,1} = \||W^{L+1}|\cdots|W^1|\|_{1,1}$ (with entry wise absolute values on all $W^l$) is an upper bound on the number of input-output dependencies, i.e., for each output $i$ summing up all the inputs $j$ it is not constant in. Regularizing input gradients [42, 41, 43] is another alternative to train neural networks that depend on fewer inputs, however it is not scalable to high-dimensional regression tasks.

---

[3]The $1, 1$ norm $\|W\|_{1,1}$ is the sum of absolute values of all entries of $W$.

Bellot et al. [7] instead train a separate neural net $f_{\theta_i} : \mathbb{R}^n \to \mathbb{R}$ for each variable $X_i$ and penalize NODEs using GroupLasso on the inputs via

$$\sum_{k,i=1}^{n} \|[W_i^1]_{\cdot,k}\|_2 \tag{3}$$

where $W_i^1$ is the weight matrix in the input layer of $f_i$ and $[W_i^1]_{\cdot,k}$ refers to the $k^{th}$ column of $W_i^1$ that should simultaneously (as a group) be set to zero or not. While enforcing strict feature sparsity (instead of regularizing an upper bound), parallel training of multiple NODE networks can be computationally expensive (Figure 1, bottom). While this work suggests that sparsity of causal interactions helps system identification, its empirical evaluation predominantly focuses on synthetic data settings, leaving performance on real data underexplored.

Another recent work suggests that standard weight or neuron pruning improves generalization for NODEs [28]. The authors show that pruning lowers empirical risk in density estimation tasks and decreases Hessian's eigenvalues, thus obtaining better generalization (flat minima). However, the effect of sparsity on identifying the underlying dynamics as well as the generalization properties of NODEs to forecast future values are not assessed.

## 3 Sparsification of neural ODEs

In an attempt to combine the strengths of both weight and feature sparsity, we propose a new regularization technique, called PathReg, which compares favorably to both C-NODE [3] and GroupLasso [7]. Before introducing PathReg, we describe how to extend existing methods to NODEs for an exhaustive empirical evaluation.

### 3.1 Methods

**L0 regularization.** Inspired by [31], we use a *differentiable* L0 norm regularization method that can be incorporated in the objective function and optimized via stochastic gradient descent. The L0 regularizer prunes the network during training by encouraging weights to get *exactly zero* using a set of non-negative stochastic gates $z$. For an efficient gradient-based optimization, Louizos et al. [31] propose to use a continuous random variable $s$ with distribution $q(s)$ and parameters $\phi$, where $z$ is then given by

$$s \sim q(s \mid \phi), \qquad z = \min(1, \max(0, s)). \tag{4}$$

Gate $z$ is a hard-sigmoid rectification of $s^2$ that allows the gate to be *exactly* zero. While we have the freedom to choose any smoothing distribution $q(s)$, we use binary concrete distribution [33, 21] as suggested by the original work [31]. The regularization term is then defined as the probability of $s$ being positive

$$q(z \neq 0 \mid \phi) = 1 - Q(s \leq 0 \mid \phi), \tag{5}$$

where $Q$ is the cumulative distribution function of $s$. Minimizing the regularizer pushes many gates to be zero, which implies weight sparsity as these gates are multiplied with model weights. L0 regularization can be added directly to the NODE network $f_\theta$.

**LassoNet** is a feature selection method [26] that uses an input-to-output skip (residual) connection that allows a feature to participate in a hidden unit only if its skip connection is active. LassoNet can be thought of as a residual feed-forward neural net $Y = \mathcal{S}^T X + h_\theta(X)$, where $h_\theta$ denotes another feed-forward network with parameters $\theta$, $\mathcal{S} \in \mathbb{R}^{n \times n}$ refers to the weights in the residual layer, and $Y$ are the responses. To enforce feature sparsity, L1 regularization is applied to the weights of the skip connection, defined as $\|\mathcal{S}\|_{1,1}$ (where $\|\cdot\|_{1,1}$ denotes the element-wise L1 norm). A constraint term with factor $\rho$ is then added to the objective to budget the non-linearity involved for feature $k$ relative to the importance of $X_k$

$$\min_{\theta, \mathcal{S}} \ \mathcal{L}_E(\theta, \mathcal{S}) + \lambda \|\mathcal{S}\|_1 \qquad \text{subject to} \ \ \|[W^1]_{\cdot,k}\|_\infty \leq \rho \|[\mathcal{S}]_{\cdot,k}\|_2 \text{ for } k \in \{1, \ldots, n\}, \tag{6}$$

where $[W^1]_{\cdot,k}$ denotes the $k^{th}$ column of the first layer weights of $h_\theta$, and $[\mathcal{S}]_{\cdot,k}$ represents the $k^{th}$ column of $\mathcal{S}$. When $\rho = 0$, only the skip connection remains (standard linear Lasso), while $\rho \to \infty$ corresponds to unregularized network training.

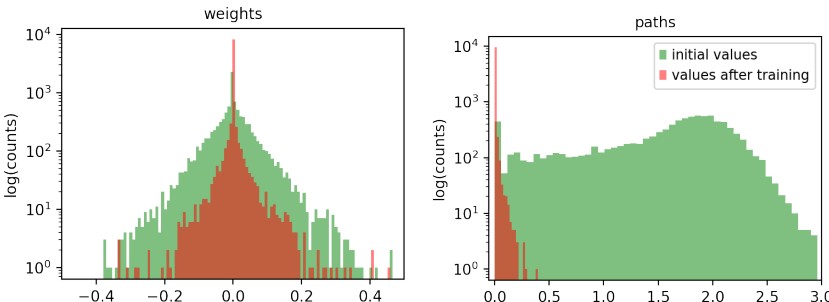

Figure 2: The distributions of network weights and paths weights (over the entries of the matrix in Eq. 13) using PathReg applied to single-cell data in Section 4.3. PathReg increases both model and feature sparsity.

LassoNet regularization can be extended to NODEs by adding the skip connection either before or after the integration (the ODE solver). If added before the integration, which we call Inner LassoNet, a linear function of $X$ is added to its time derivative

$$\dot{X} = \mathcal{S}^T X + f_\theta(X, t), \quad X(0) = x_0 . \tag{7}$$

Adding the skip connection after the integration (and the predictor $o$), called Outer LassoNet, yields

$$X_t = \mathcal{S}^T X_t + o(\text{ODESolver}(f_\theta, x_0, t_0, t)) . \tag{8}$$

**PathReg (ours).** While L0 regularization minimizes the number of non-zero weights in a network, it does not necessarily lead to feature sparsity meaning that the response variables can still depend on all features including spurious ones. To constrain the number of *input-output paths*, i.e., to enforce feature-sparsity, we regularize the probability of any path throughout the entire network contributing to an output. To this end, we use non-negative stochastic gates $z = g(s)$ similar to Eq. 4, where the probability of an input-output path $\mathcal{P}$ being non-zero is given by

$$q(\mathcal{P} \neq 0) = \prod_{z \in \mathcal{P}} q(z \neq 0 \mid \phi) \tag{9}$$

and we constrain

$$\sum_{i=1}^{\#paths} \prod_{z \in \mathcal{P}_i} q(z \neq 0 \mid \phi) \tag{10}$$

to minimize the number of paths that yield non-zero contributions from inputs to outputs. This is equivalent to regularizing the *gate adjacency matrix* $A_z = G^{L+1} \cdot \ldots \cdot G^1$. Where $G^l$ is a probability matrix corresponding to the probability of the $l$-th layer gates being positive. Then, $A_{z_{ij}}$ represents the sum of the probabilities of all paths between the $i$-th input and the $j$-th output. Ultimately, we thus obtain our PathReg regularization term

$$\|A_z\|_{1,1} = \|G^{L+1} \ldots G^1\|_{1,1} \quad \text{with} \quad G_{ij}^l = q^l(z_{ij} \neq 0 \mid \phi_{i,j}) , \tag{11}$$

where $q^l(z_{ij} \neq 0)$ with parameters $\phi_{i,j}$ is the probability of the $l$-th layer gate $z_{ij}$ being nonzero. Regularizing $\|A_z\|_{1,1}$, minimizes the number of paths between inputs and outputs and induces no shrinkage on the actual values of the weights. Therefore, we can utilize other regularizers on $\theta$ such as $\|A\|_{1,1}$ in conjunction with PathReg similar to Eq. 2. In this work, we consider the following overall loss function

$$\mathcal{R}(\theta, \phi) = \mathbb{E}_{q(s|\phi)} \left[ \frac{1}{N} \left( \sum_{i=1}^{N} \mathcal{L}\big(o(f(x_i; \theta \odot g(s)), y_i)\big) \right) \right] + \lambda_0 \|A_z\|_{1,1} + \lambda_1 \|A\|_{1,1}$$

$$= \mathcal{L}_E(\theta, \phi) + \lambda_0 \|A_z\|_{1,1} + \lambda_1 \|A\|_{1,1} \tag{12}$$

where $\mathcal{L}$ corresponds to the loss function of the original predictive task, $\mathcal{L}_E$ the overall predictive loss, $o$ is the NODE-based model with $f$ modeling time derivatives, $g(\cdot) := \min(1, \max(0, \cdot))$, $\lambda_0, \lambda_1 \geq 0$ are regularization parameters, and $\odot$ is entry-wise multiplication. $\mathcal{L}_E$ measures how well the model

Table 1: Results for the synthetic second-order ODE.

| MODEL | MSE | | SPARSITY (%) | |
|---|---|---|---|---|
| | TRAIN | EXTRAPOLATION | FEATURES | WEIGHTS |
| BASE | $1.3 \times 10^{-4}$ | $3.2 \times 10^{-3}$ | 47.9 | 11.1 |
| L0 | $1.9 \times 10^{-2}$ | $3.3 \times 10^{-2}$ | 0.0 | **49.6** |
| C-NODE | $6.6 \times 10^{-5}$ | $4.1 \times 10^{-4}$ | **75.5** | 16.7 |
| PATHREG | $\mathbf{6.3 \times 10^{-5}}$ | $\mathbf{3.9 \times 10^{-4}}$ | 75.5 | 35.2 |
| LASSONET | $8.4 \times 10^{-3}$ | $4.0 \times 10^{-2}$ | 0.0 | 0.0 |
| GROUPLASSO | $8.8 \times 10^{-5}$ | $1.5 \times 10^{-3}$ | 61.1 | 6.0 |

fits the current dataset. While $\|A\|_{1,1}$ shrinks the actual values of weights $\theta$ in an attempt to zero out entire paths, $\|A_z\|_{1,1}$ enforces exact zeros for entire paths at once. When $A_{z_{ij}} = 0$ the output $i$ is constant in input $j$. In practice, during training the expectation in Eq. 12 is estimated as usual via Monte Carlo sampling.

Unlike GroupLasso [7], PathReg does not require training of multiple networks in parallel. Moreover, unlike C-NODE [3], PathReg leads to exact zeros in the weight matrices and requires no choice of threshold for deciding which paths are considered as zeros (see Figure 2).

### 3.2 Training sparse models

Sparsity can generally be enforced during or after model training. In this paper, we study two main approaches [20] to train sparse models: (i) *sparse training*, where we add regularizers already during model training; (II) *training and iterative sparsification*, where we iteratively increase regularization parameters only after the model has been trained (close to) to convergence. A detailed description of our training procedures and architecture choices for each experiment is provided in Appendix A.

### 3.3 Assessing feature sparsity

For models regularized with L0 or PathReg the natural measure for feature sparsity is

$$(|W^{L+1}| \odot G^{L+1}) \cdot \ldots \cdot (|W^1| \odot G^1) , \tag{13}$$

where absolute values are taken entry-wise. LassoNet also allows for a natural feature sparsity metric by counting the non-zero input weights $\sum_{j=1}^{n} \|W_j^{(1)}\|_2 > \epsilon$ of the first hidden layer, where $n$ denotes the number of input features and $\epsilon$ is a small tolerance parameter. To evaluate feature sparsity for other feed-forward continuous-depth models, we count the number of non-zero elements in $A = |W^{L+1}| \cdots |W^1|$ again with entry wise absolute values on all $W$ and within some tolerance $\epsilon$ (set to $1e - 5$ in our experiments). This amounts to an upper bound on the number of input-output dependencies: while we can have $A_{i,j} \neq 0$ even though $X_j \notin pa_{f_\theta}(X_i)$, $A_{i,j} = 0$ always implies $X_j \notin pa_{f_\theta}(X_i)$.

## 4 Results

In this section, we present empirical results on several datasets from different domains including human motion capture data and large-scale single-cell RNA-seq data. We demonstrate the effect of the different sparsity methods including L0 [31], C-NODE [3], LassoNet [26], GroupLasso [7], and our PathReg on the generalization capability of continuous-depth neural nets. We address different aspects of generalization by assessing the accuracy of these models for time-series forecasting and the identification of the governing dynamical laws. In all examples, we assume that the observed motion can be (approximately) described by a system of ODEs where variables correspond directly to the observables of the trajectories.

### 4.1 Sparsity for system identification

We motivate sparsity for NODEs starting with a simple second-order NODE as used by Aliee et al. [3]. The differential equations explaining this system are presented in Appendix B.1. The questions

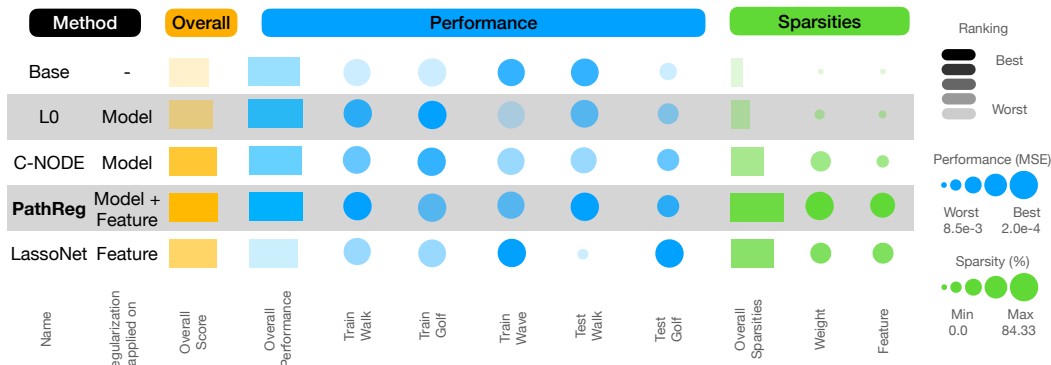

Figure 3: Qualitative result comparison for the human motion capture data. Circle sizes indicate the scaled (continuous) metrics: MSE (blue) and sparsity (in %) (green). The lengths of bar show the mean metric over all sparsities / performances for a method. Different shades of the same color indicate the ranking from best to worst method within a given column. The overall score (orange) is the average over all the scores, including both performance and sparsity, for each method across datasets. We then rank the methods accordingly. PathReg yields the best overall results in this comparison.

that we try to answer here are whether different sparsity methods can (i) learn the true underlying system, and (ii) forecast into the future (extrapolate). For this linear system, the adjacency matrix represents the exact coefficients in Eq. 14. The results summarized in Tables 1 and 2 show that PathReg, C-NODE, LassoNet, and GroupLasso perform well with respect to reconstruction error. Moreover, PathReg, C-NODE, and GroupLasso lead to higher sparsity and outperform other methods for extrapolation. Among these methods, only PathReg and C-NODE are able to learn the true system with the lowest error[4].

## 4.2 Sparsity improves time-series forecasting

We next demonstrate the robustness of sparse models for both reconstructing and extrapolating human movements using real motion capture data from `mocap.cs.cmu.edu`. Each datapoint is 93-dimensional (31 joint locations with three dimensions each) captured over time. We select three different movements including walking, waving, and golfing, and use 100 frames each for training. After training, we query all models to extrapolate the next 100 frames. For walking and golfing, where multiple trials are available, we also test the model on unseen data in the sense that it has not seen any subset of those specific sequences (despite having trained on other sequences depicting the same type of movement).

Generally, for time-series forecasting, it is not straight forward to give precise definitions of in-, and out-of-distribution. In Table 7 in Appendix C, we show a grid of plots illustrating loss and sparsity as a function of the regularization parameter $\lambda$ of each method on this dataset. We show detailed results in Table 3 in Appendix C and summarize them concisely in Figure 3. We did not manage to scale GroupLasso to this 93-dimensional dataset. While all other models show comparative performance (in terms of MSE error), only PathReg achieves strong levels of both weight and feature sparsity (quantitative comparison is in Table 3). The results show that PathReg outperforms other meth-

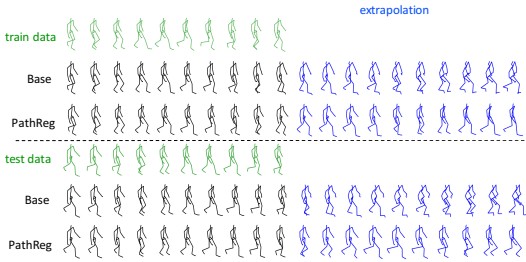

Figure 4: Human-movement reconstructions and extrapolation.

---

[4]GroupLasso is originally designed for first-order ODEs. Here, we use an augmented version of that feeding also the initial velocities to the network. Therefore, a direct comparison with its adjacency matrix is not entirely adequate.

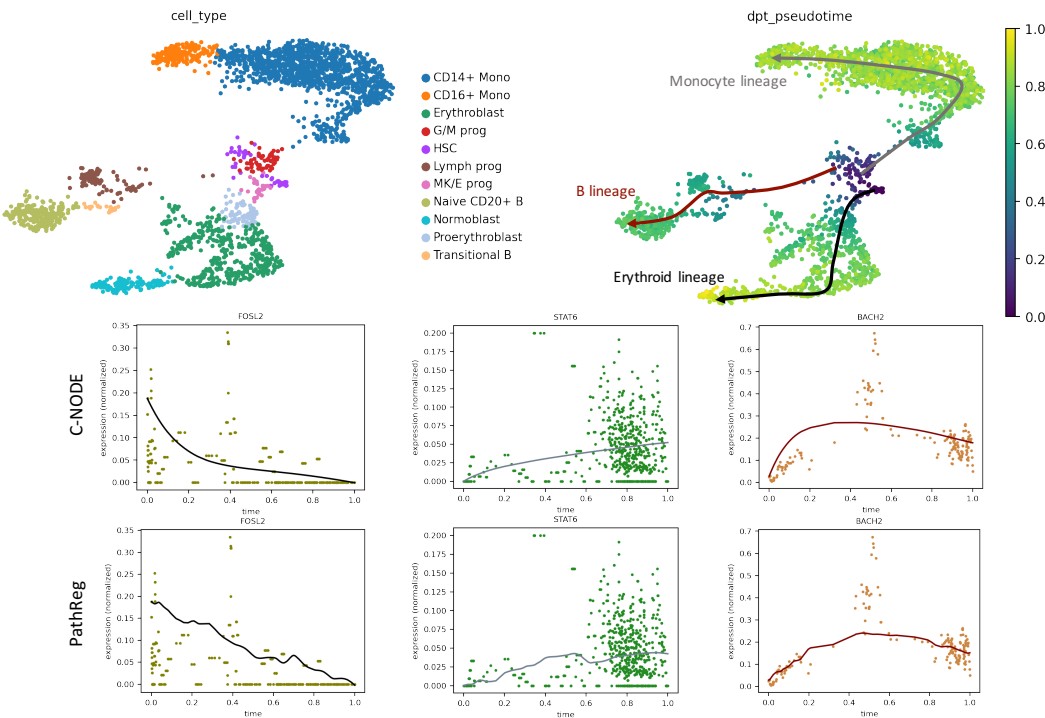

Figure 5: Single-cell RNA-seq Data. Shown are 2D UMAP visualisation of training data colored by cell types and pseudotime (top) where each point is a cell, and the expressions of three genes and their predictions over pseudo time using C-NODE (top) and PathReg (bottom). The predictions across all genes are presented in Figure 9.

ods with respect to MSE for both train and test datasets while resulting in the highest model and feature sparsity.

We further examine the extrapolation (in time) capability of these models. We observe that all sparsity regularization techniques improve extrapolation over unregularized NODEs. Figure 4 qualitatively illustrates the reconstruction and extrapolation performance of both the baseline and PathReg for the walking dataset (visually, the results are similar to PathReg for other regularizers). While these models perform well for extrapolation and generalization to unseen data, we show that only PathReg and C-NODE are able to avoid using spurious features for forecasting. Figures 7 and 8 in Appendix C show that PathReg and C-NODE learn sensible and arguably correct adjacency matrices (derived from joint connectedness), whereas L0 and LassoNet fail to sparsify the input-output interactions appropriately (more details in Appendix C.2). Since we use second-order NODEs for this dataset, the adjacency matrix described in Section 3.3 represents how the acceleration predicted by $f_\theta$ of each joint point (e.g., $X_i$) depends on the positions as well as the velocities of other joint points (e.g., $X_j$). In case the dependency is non-constant, we infer that the joint point $X_i$ interacts with $X_j$.

### 4.3 Sparsity improves dynamic identification for single-cell data

Finally, we explore how pruning (i) improves real-world generalization and (ii) helps to learn gene-gene interactions in human hematopoiesis single-cell multiomics data [32] (GEO accession code = GSE194122). The regulatory (or causal) dependencies between genes can explain cellular functions and help understanding biological mechanisms [1]. Gene interactions are often represented as a gene regulatory network (GRN) where nodes correspond to genes and directed edges indicate regulatory interactions between genes. Reliable GRN inference from observational data is exceptionally challenging and still an open problem in genomics [40]. As human regulatory genes known as transcription factors ($\sim$ 1,600 [25]), do not individually target all genes (potentially up to $\sim$ 20,000), the inferred GRN is expected to be *sparse*. Thus, predicting the expression of a gene should depend on few other genes that can be restricted using sparsity-enforcing regularizers.

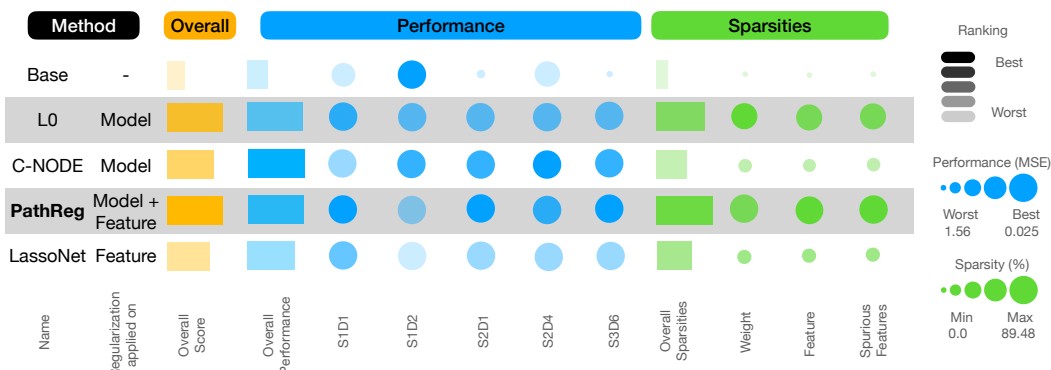

Figure 6: Comparison results using the single-cell RNA-seq Data. PathReg shows best overall results, as well as with regard to performance, and to sparsity. The models are trained with batch S1D2, and tested using other four batches.

For assessing real-world generalization of various sparsity-enforcing methods, we use human bone marrow data of four diverse donors, generated at three different sites using five sequencing batches (the pre-processing steps applied to the data are discussed in Appendix B.3). In this experiment, we train NODE models with one batch (S1D2), and later test those using other four batches (S1D1, S2D1, S2D4, and S3D6). Notably, time-resolved ground truth dynamics of cells throughout their lifetime are not available, because all existing genome-wide technologies are destructive to the cells [32]. Therefore, scRNA-seq only offers static snapshots of data and the temporal order of cell states is commonly estimated based on certain distances between their measured transcriptomes. Following that, we first apply the diffusion pseudo-time (DPT) ordering approach [19], a diffusion map based manifold learning technique, to rank the cells. This allows to generate a cell fate mapping from hematopoietic stem cells (HSC) into three biological lineages that are used for training: Monocyte, Erythroid, and B cells (Figure 5). NODEs are then trained using highly-variable genes (529 genes) with time taken to be the inferred pseudo-time (Figure 9).

The train and test MSE across 5 batches and three pruning methods are presented in Table 5 and Table 7 in Appendix C.3 and summarized in Figure 6. While all regularized models achieve decent predictive performance, PathReg outperforms the other methods in most scenarios. We further study the transcription-factor-to-target-gene interactions via the adjacency matrices filtered for transcription factors (columns) and target genes (rows) (see Appendix B.3 for more details). The adjacency matrix $A$ demonstrates what transcription factors (on columns) are used for predicting the expression value of a target gene (on rows). $|A_{ij}| > \epsilon$ for a small threshold $\epsilon = 1e - 5$ can then be interpreted as an interaction between gene $i$ and gene $j$ being present in the data (see Figure 10 for an illustrating example). For visualisation purposes, Figure 11 only shows the adjacency matrices across the Erythroid lineage. However, the results are comparable for other lineages. The used list of transcription factors and target genes are extracted using chromatin and sequence-specific motif information (we again refer to Appendix B.3 for more details). Since ground truth dynamics are not known for GRNs, validating the inferred interactions for multiple cell types is a challenging task. However, we observe that PathReg and L0 are able to remove the interactions between pairs known to not directly causally influence each other, including target genes (called spurious features in Table 5 and further discussed in Appendix B.3), while the other models fail in that regard. Assessed by the literature, we also observe that known Erythroid transcription factors are inferred as important by both PathReg and L0 (RUNX1 [24], ETV6 [49], NFIA [34], MAZ [12], GATA1 [53]).

## 5 Conclusion and discussion

We have shown the efficacy of various sparsity enforcing techniques on generalization of continuous-depth NODEs and proposed PathReg, a regularizer acting directly on entire paths throughout a neural network and achieving exact zeros. We curate real-world datasets from different domains consisting of human motion capture and single-cell RNA-seq data. Our findings demonstrate that sparsity improves out-of-distribution generalization (for the types of OOD considered) of NODEs

when our objective is system identification from observational data, extrapolation of a trajectory, or real-world generalization to unseen biological datasets collected from diverse donors. Finally, we show that unlike weight sparsity, feature sparsity as enforced by PathReg can indeed help in identifying the underlying dynamical laws instead of merely achieving strong in-distribution predictive performance. This is particularly relevant for applications like gene-regulatory network inference, where the ultimate goal is not prediction and forecasting, but revealing the true underlying regulatory interactions between genes of interest. We hope that our empirical findings as well as curated datasets can serve as useful benchmarks to a broader community and expect that extensions of our framework to incorporate both observational and experimental data will further improve practical system identification for ODEs. Finally, our path-based regularization technique may be of interest to other communities that aim at enforcing various types of shape constraints or allowed dependencies into deep learning based models[5].

## Acknowledgments and Disclosure of Funding

We thank Ronan Le Gleut from the Core Facility Statistical Consulting at Helmholtz Munich for the statistical support. HA and FJT acknowledge support by the BMBF (grant 01IS18036B) and the Helmholtz Association's Initiative and Networking Fund through Helmholtz AI (grant ZT-I-PF-5-01) and CausalCellDynamics (grant Interlabs-0029). TR is supported by the Helmholtz Association under the joint research school "Munich School for Data Science" - MUDS. NK was funded by Helmholtz Association's Initiative and Networking Fund through Helmholtz AI. FJT reports receiving consulting fees from Roche Diagnostics GmbH and ImmunAI, and ownership interest in Cellarity, Inc. and Dermagnostix. The remaining authors declare no competing interests.

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
