# A    Models

All neural networks used in this work are fully connected, feed-forward neural networks. First-order NODEs are used for single-cell data, while second NODEs are used for the synthetic example as well as the motion capture data. In the second-order NODEs, the initial velocities are predicted using a neural network with two hidden layers with 20 or 100 neurons depending on the dataset with ELU activation function. The main architecture to infer velocities (or accelerations) also contains two hidden layers of sizes 20 or 100 depending on the size of the input and ELU activation function. As an ODE solver, we use an explicit 5-th order Dormand-Prince solver commonly denoted by `dopri5`.

All models are optimized using Adam [23] with an initial learning rate of $0.01$. We use PyTorch's default weight initialization scheme for the weights. All models can be trained entirely on CPUs on consumer grade Laptop machines within minutes or hours. Execution times per epoch for the single-cell data with 529 features are as follows: Base=0.9, L0 = 1.3, L1 = 0.95, PathReg = 1.05, GroupLasso = (not scalable), LassoNet = 0.83 seconds.

Except LassoNet that iteratively increases the sparsity regularization parameter, we observe that the rest of the studied methods perform better when the regularization term is fixed during the training (sparse training).

# B    Datasets

### B.1    Synthetic second-order ODE

We use a second-order, homogeneous, linear system of ODEs with constant coefficients to assess how pruning affects system identification using NODEs. Our system is denoted as:

$$X_0 = -1.5 \cdot X_0 - 0.5 \cdot \dot{X}_0 \tag{14}$$

$$X_1 = -2.7 \cdot X_1 - 1.0 \cdot \dot{X}_1 \tag{15}$$

$$X_2 = -2.0 \cdot X_2 - 1.0 \cdot \dot{X}_0 - 1.0 \cdot \dot{X}_1 - 0.1 \cdot \dot{X}_2 \tag{16}$$

We also set $X(0) = (0.4, 0.7, 1.8)$ and $\dot{X}(0) = (0.3, 0.5, 1.0)$ as initial positions and velocities, respectively.

### B.2    Motion capture data

The Carnegie Mellon University Motion Capture Database [46] is a large collection of human motion data. We are only interested in those records where activities are performed by a single person who does not interact with other people and has minimal contact with objects within the environment.The data is presented as 31 joint points in 3D space (in total of 93 variables) and a graph links them. The data is preprocessed as the following:

- **Centering the frames:** The root point is fixed at the origin $(0, 0, 0)$ for the whole duration of the movement. We observe that this method is effective particularly for walking data.
- **Centering the first frame:** For golfing and waving, the root point of the first frame is moved to the origin $(0, 0, 0)$.
- **Setting the floor at $z = 0$:** We find the minimum value of the Z-coordinate throughout the trial and among all joint points. We consider this level to be the "floor" and move all points so that the minimum value of $z$ during the whole trial is at 0 on the Z-axis.

### B.3    Single-Cell RNA-seq

The single-cell RNA count data is first normalized for library size to remove any difference that is arisen due to sampling effects. Gene counts across lineages are then scaled to improve comparisons between genes by weighting them equally. While we expect batch correction to help our method in that it should be easier to predict dynamics for unseen batches, we intentionally avoided that, interpreting each batch as some form of OOD experiment to study the generalization of the methods.

To map putative transcription factor (TF) and target gene relationships, we use as a reference a regulatory network generated using the gene expression and chromatin accessibility features

available in the human immune cells dataset. Briefly, using the function *rank_genes_groups* from SCANPY [50], we select the top 1,000 genes and 25,000 chromatin peaks, using cell type label as a covariate. Then, for each chromatin peak we map sequence-specific TF motifs using as reference the TF archetypes from ENCODE (n=236), Peaks are mapped to the closest gene using the R package ChIPseeker, with default parameters for the genome build hg38 [47]. All chromatin peaks linked to a gene as an enhancer annotation are considered, as long as the mapped distance between peak and promoter is below 200 Kbp. Promoter/intron/exon peaks are always mapped to the harboring gene. Our rule for successfully mapping a TF to a target gene through a chromatin peak is that all TF, chromatin peak, and target gene, have to be simultaneously in the list of features selected in the *rank_genes_groups* function for cell type of interest, and there have to be TF motifs linked to that transcription factor in the chromatin peak. This gives us a total of 113 TFs and 365 potential targets genes across all cell types, that we use for evaluating the models. These curated relationships between TFs and target gene are preliminarily useful for assessing if our model captures associations that can have some level of biological support. That means, if $f$ indicates a dependency between a TF and a target gene, being also recovered in this list, we can suggest that this relationship is supported by the existence of a TF-peak-gene association. We also define the interactions between target genes not linked to TFs through peaks as *spurious*, given their lack of biological support for direct gene regulation. Of course, we emphasize that our choice of "spurious" might not certainly capture all types of spurious associations. Curating a list of cell-type specific TFs might offer additional, context-specific support, but only for a handful of TFs.

## C   Extended results

### C.1   Synthetic second-order ODE

Table 2 shows the inferred adjacency matrices, reconstructed and extrapolated samples. The values in the adjacency matrices show the coefficients of the learned differential equations with the true coefficients presented in Eq. 14, 15,and 16. Only C-NODE and PathReg are able to learn the true dynamics.

### C.2   Motion capture data

Tables 3 and 4 show train and test MSE as well as model and feature sparsity for motion capture data. The inferred adjacency matrix for walking motion using PathReg is illustrated in Figure 7. The joint points corresponding to each leg or hand are highlighted with dashed lines. The extended results across different methods and motions are also presented in Figure 8. We observe that only PathReg and C-NODE are able to learn the important interactions. For example, when waving with the left hand, PathReg correctly identifies only interactions between the joint points corresponding to the left arm. For the golfing movement, PathReg correctly recovers that both arms move in a connected fashion, but legs and most other body parts are not substantially involved in the movement. However, the base model with no regularizer and L0 fail to sparsify the interactions. While LassoNet can sparsify the interactions, we do not observe a meaningful association with the important joint points.

### C.3   Single-cell data

Tables 5 and 6 compare MSE and sparsity of various models. Figure 9 presents the real and predicted expression of genes over time for three lineages including Monocyte, Erythroid, and B using PathReg. The inferred adjacency matrices are also illustrated in Figure 11.

### C.4   Analysis of the effect of regularization on sparsity and performance

The plots in Table 7 show the effects that different regularization weights have on performance and model as well as feature sparsity.

Table 2: Results of the synthetic second-order ODE.

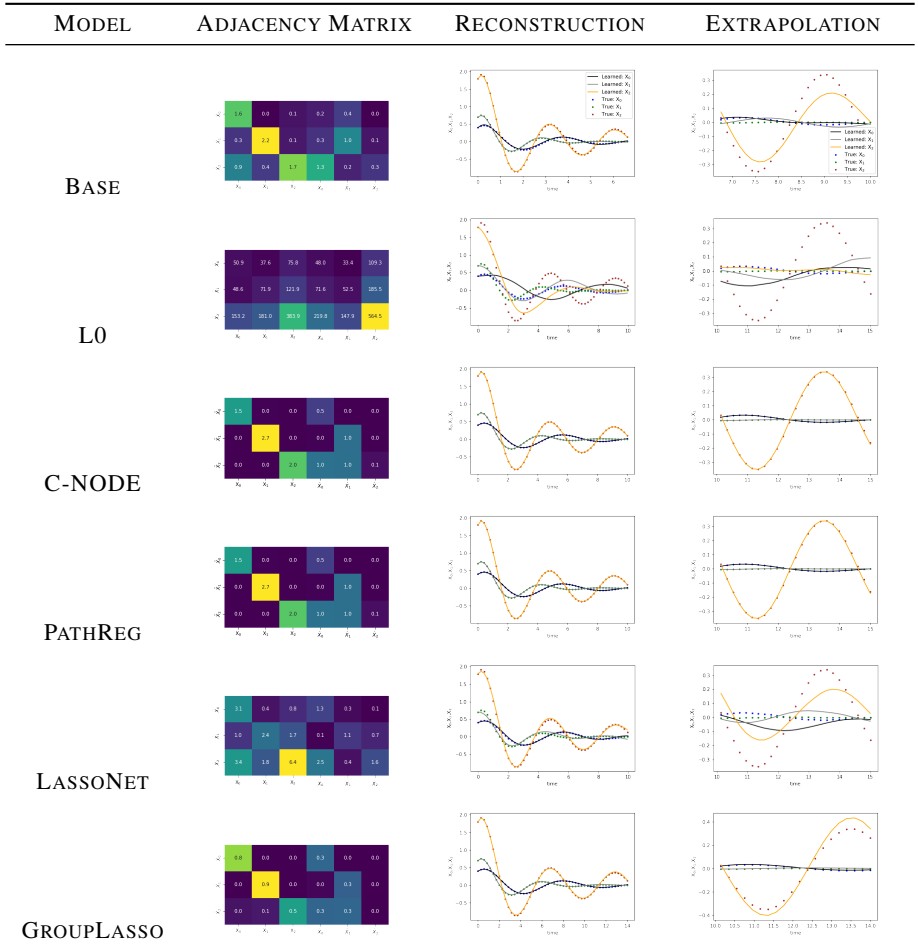

Table 3: Train and test MSE for human motion capture data. Shown are average feature and parameter sparsity for three models trained with different motions (walk, wave, and golf). Outer LassoNet is presented in this benchmark, see Table 4 for comparison to Inner LassoNet.

| | | BASE | L0 | C-NODE | PATHREG | LASSONET |
|---|---|---|---|---|---|---|
| TRAIN MSE | WALK | $7.0 \times 10^{-4} \pm 9.4 \times 10^{-5}$ | $3.1 \times 10^{-4} \pm 2.0 \times 10^{-5}$ | $6.0 \times 10^{-4} \pm 2.0 \times 10^{-4}$ | $\mathbf{2.9 \times 10^{-4} \pm 1.8 \times 10^{-4}}$ | $7.2 \times 10^{-4} \pm 2.9 \times 10^{-6}$ |
| | WAVE | $2.4 \times 10^{-4} \pm 2.3 \times 10^{-6}$ | $\mathbf{2.0 \times 10^{-4} \pm 1.5 \times 10^{-7}}$ | $2.0 \times 10^{-4} \pm 2.1 \times 10^{-5}$ | $2.0 \times 10^{-4} \pm 4.8 \times 10^{-4}$ | $5.1 \times 10^{-4} \pm 1.6 \times 10^{-5}$ |
| | GOLF | $6.6 \times 10^{-4} \pm 1.5 \times 10^{-2}$ | $7.0 \times 10^{-4} \pm 4.2 \times 10^{-4}$ | $6.9 \times 10^{-4} \pm 4.3 \times 10^{-4}$ | $6.8 \times 10^{-4} \pm 4.0 \times 10^{-5}$ | $\mathbf{3.3 \times 10^{-4} \pm 3.3 \times 10^{-5}}$ |
| TEST MSE | WALK | $2.4 \times 10^{-3} \pm 2.1 \times 10^{-4}$ | $2.3 \times 10^{-3} \pm 3.1 \times 10^{-2}$ | $2.7 \times 10^{-3} \pm 1.2 \times 10^{-4}$ | $\mathbf{2.0 \times 10^{-3} \pm 2.0 \times 10^{-4}}$ | $8.5 \times 10^{-3} \pm 1.3 \times 10^{-3}$ |
| | GOLF | $4.5 \times 10^{-3} \pm 2.0 \times 10^{-2}$ | $3.1 \times 10^{-3} \pm 1.1 \times 10^{-5}$ | $2.8 \times 10^{-3} \pm 4.3 \times 10^{-4}$ | $2.7 \times 10^{-3} \pm 1.7 \times 10^{-4}$ | $\mathbf{4.8 \times 10^{-4} \pm 3.1 \times 10^{-4}}$ |
| FEAT. SPAR. (%) | | $0.0 \pm 0.0$ | $8.06 \pm 2.04$ | $25.33 \pm 2.41$ | $\mathbf{71.33 \pm 2.19}$ | $57.48 \pm 47.21$ |
| MODEL SPAR. (%) | | $0.0 \pm 0.0$ | $17.11 \pm 1.10$ | $55.66 \pm 1.93$ | $\mathbf{84.33 \pm 2.51}$ | $57.42 \pm 47.20$ |

Table 4: Comparison of Inner- and Outer LassoNet. Train and test MSE for human motion capture data. Shown are average feature and parameter sparsity for three models trained with different motions (walk, wave, and golf). Outer LassoNet shows higher sparsity and better performance except for the walking test data and is therefore included in Table 3.

|  |  | INNER LASSONET | OUTER LASSONET |
|---|---|---|---|
| TRAIN MSE | WALK | $7.5 \times 10^{-4} \pm 4.3 \times 10^{-6}$ | $\mathbf{7.2 \times 10^{-4} \pm 2.9 \times 10^{-6}}$ |
|  | WAVE | $6.3 \times 10^{-4} \pm 1.6 \times 10^{-5}$ | $\mathbf{5.1 \times 10^{-4} \pm 1.6 \times 10^{-5}}$ |
|  | GOLF | $7.3 \times 10^{-4} \pm 2.0 \times 10^{-5}$ | $\mathbf{3.3 \times 10^{-4} \pm 3.3 \times 10^{-5}}$ |
| TEST MSE | WALK | $\mathbf{1.2 \times 10^{-3} \pm 4.6 \times 10^{-5}}$ | $8.5 \times 10^{-3} \pm 1.3 \times 10^{-3}$ |
|  | GOLF | $4.0 \times 10^{-3} \pm 4.6 \times 10^{-4}$ | $\mathbf{4.8 \times 10^{-4} \pm 3.1 \times 10^{-4}}$ |
| FEAT. SPAR. (%) |  | $7.01 \pm 15.66$ | $\mathbf{57.48 \pm 47.21}$ |
| MODEL SPAR. (%) |  | $6.31 \pm 15.20$ | $\mathbf{57.42 \pm 47.20}$ |

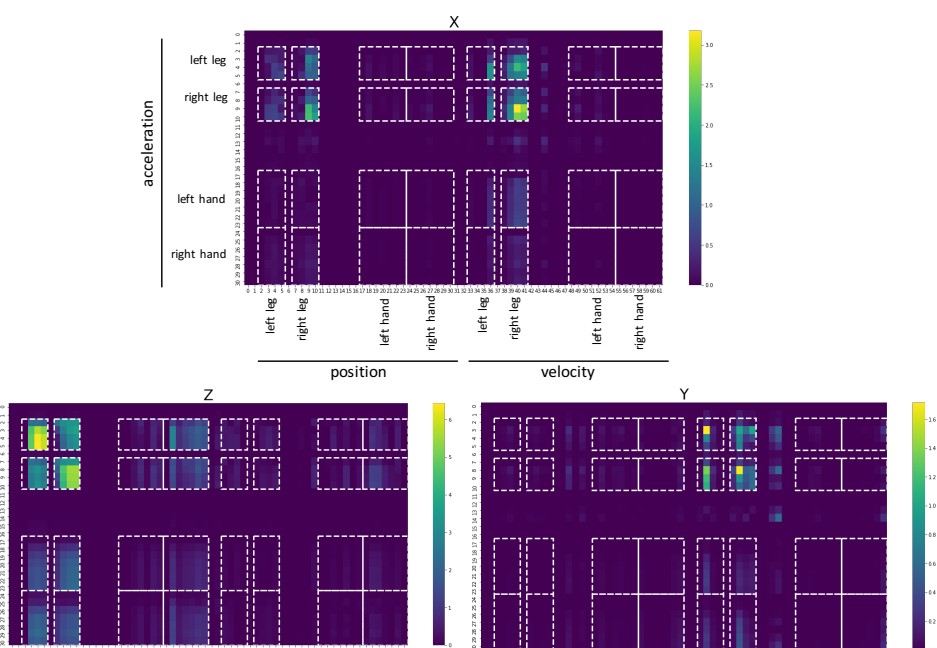

Figure 7: The adjacency matrix for MCD experiment trained with PathReg-regularized NODE. Shown are the interactions among joint points across three dimensions (X, Y, and Z). Here, a second-order NODE is trained that infers acceleration as a function of position and velocity. Then, $X_j$ interacts with $X_i$ if $X_j \in pa(X_i)$ or $\dot{X}_j \in pa(X_i)$, where $\ddot{X}_i = f(pa(X_i), t)$.

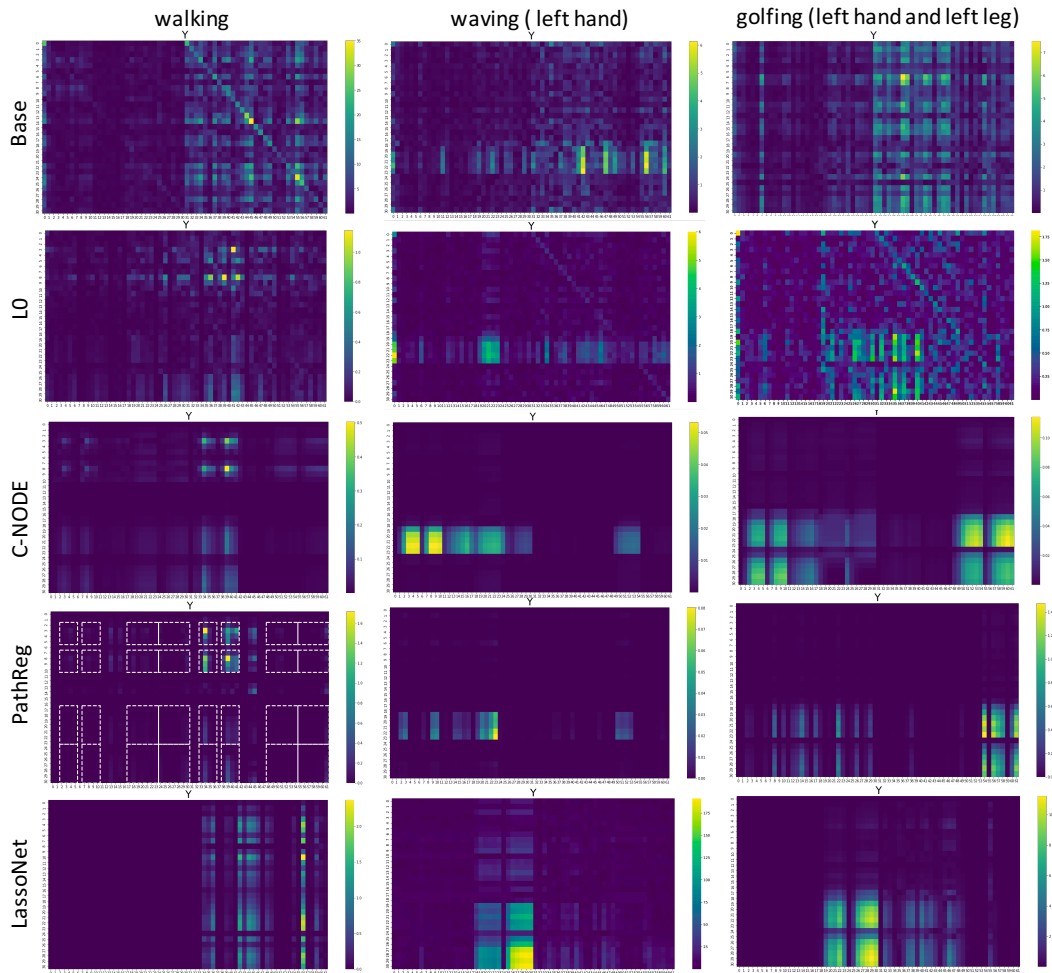

Figure 8: The adjacency matrices for MCD data trained with different pruning methods. PathReg and C-NODE find the most relevant features. For example for waving with the left hand, there are only interactions with the joint points corresponding to the left hand. For the sake of simplicity, we only show the interactions across dimension Y.

Table 5: Generalization for single-cell data. NODEs are trained with batch S1D2 and tested for four other batches. Shown are MSE as well as model and feature sparsity. Inner LassoNet is presented in this benchmark, see Table 6 for comparison to Outer LassoNet.

|  |  | BASE | L0 | C-NODE | PATHREG | LASSONET |
|---|---|---|---|---|---|---|
| MSE | S1D1 | $0.34 \pm 0.40$ | $0.038 \pm 5.5 \times 10^{-4}$ | $0.038 \pm 0.097$ | $\mathbf{0.036 \pm 2.6 \times 10^{-4}}$ | $0.044 \pm 6.4 \times 10^{-4}$ |
|  | S1D2 | $\mathbf{0.033 \pm 1.7 \times 10^{-4}}$ | $0.035 \pm 1.2 \times 10^{-4}$ | $0.033 \pm 2.4 \times 10^{-4}$ | $0.035 \pm 3.6 \times 10^{-4}$ | $0.051 \pm 3.8 \times 10^{-4}$ |
|  | S2D1 | $1.39 \pm 1.90$ | $0.042 \pm 1.0 \times 10^{-3}$ | $0.038 \pm 0.021$ | $\mathbf{0.037 \pm 4.0 \times 10^{-4}}$ | $0.046 \pm 4.6 \times 10^{-4}$ |
|  | S2D4 | $0.22 \pm 0.27$ | $0.026 \pm 4.2 \times 10^{-4}$ | $0.025 \pm 0.027$ | $\mathbf{0.025 \pm 2.9 \times 10^{-4}}$ | $0.030 \pm 2.2 \times 10^{-4}$ |
|  | S3D6 | $1.56 \pm 1.36$ | $0.075 \pm 4.2 \times 10^{-4}$ | $0.079 \pm 0.025$ | $\mathbf{0.072 \pm 3.0 \times 10^{-4}}$ | $0.081 \pm 3.4 \times 10^{-4}$ |
| FEAT. SPAR. (%) |  | $0.0 \pm 0.0$ | $79.58 \pm 0.80$ | $28.83 \pm 11.9$ | $\mathbf{88.36 \pm 0.33}$ | $32.67 \pm 41.56$ |
| MODEL SPAR. (%) |  | $0.0 \pm 0.0$ | $\mathbf{84.29 \pm 0.69}$ | $50.65 \pm 11.04$ | $81.06 \pm 0.45$ | $67.05 \pm 21.31$ |
| SPUR. FEAT. SPAR. (%) |  | $0.0 \pm 0.0$ | $81.49 \pm 0.95$ | $31.81 \pm 11.30$ | $\mathbf{89.48 \pm 0.54}$ | $33.57 \pm 42.06$ |

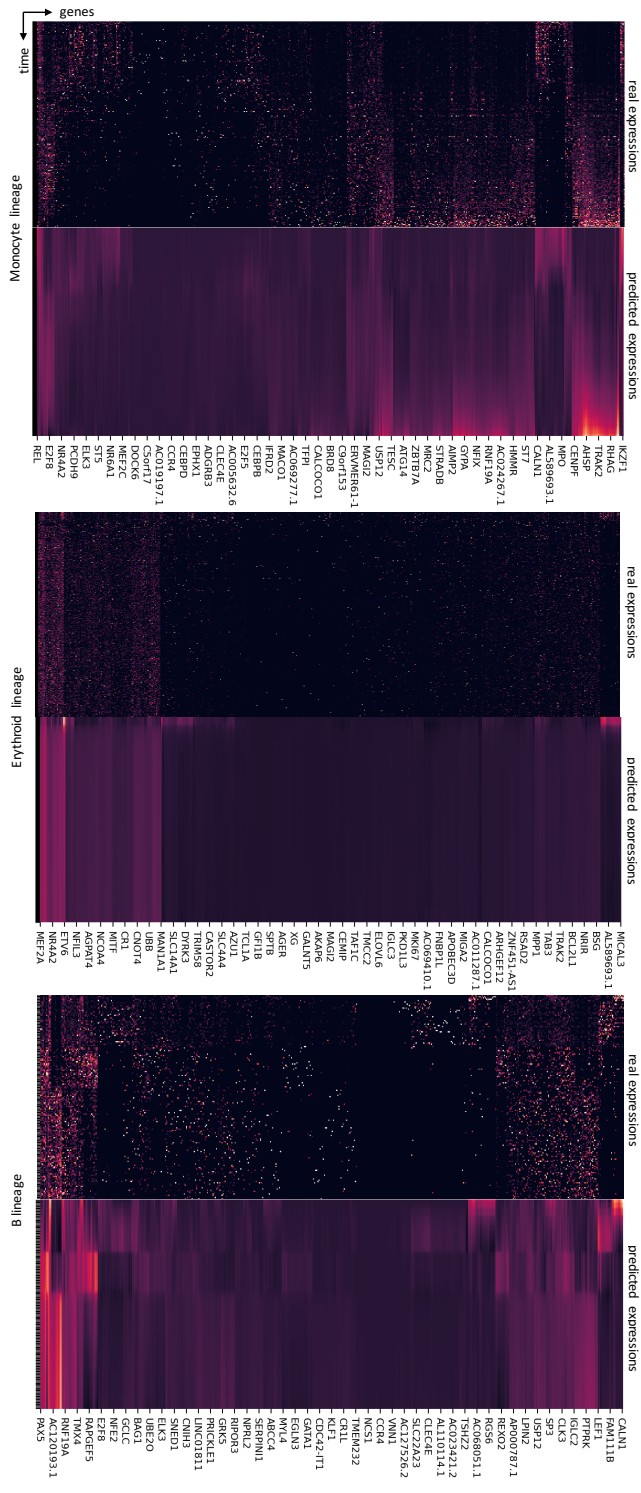

Figure 9: Normalized gene expressions over pseudotime. Shown are real (top) and predicted (bottom) expressions using PathReg.

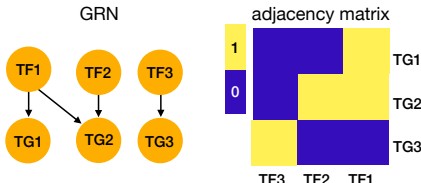

Figure 10: A GRN example compared with the true adjacency matrix (shown is a subset referring to TF-TG interactions) that we expect a model to learn.

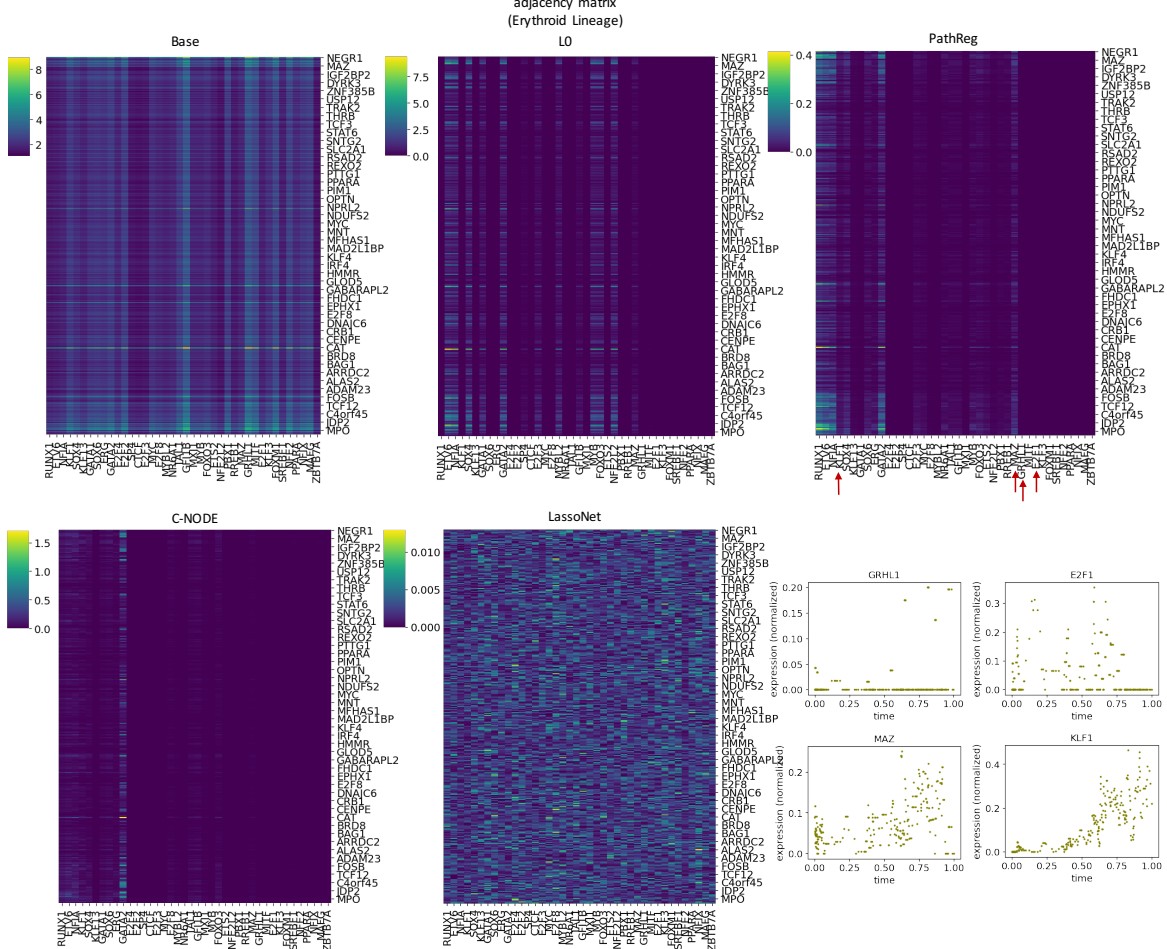

Figure 11: Adjacency matrix comparison. Shown are subsets of absolute adjacency matrices. Columns refer to transcription factors and rows refer to target genes associated with Erythroid lineage. The expression of four genes across pseudo-time are presented on right. Shown are two relevant (MAZ and KLF1) and two spurious genes (GRHL1 and E2F1). Evident by the adjacency matrices, PathReg and L0 remove the links *to* the spurious target genes or *from* the spurious transcription factors.

Table 6: Comparison Inner LassoNet and Outer LassoNet for single-cell RNA-seq data. While Outer LassoNet produces a higher performance for the in-distribution set S1D2 and higher sparsities, the generalization abilities are worse than the Inner LassoNet. Therefore, Inner LassoNet is used in the benchmark for single-cell data.

| | | INNER LASSONET | OUTER LASSONET |
|---|---|---|---|
| MSE | S1D1 | $\mathbf{0.044 \pm 6.4 \times 10^{-4}}$ | $0.16 \pm 3.9 \times 10^{-2}$ |
| | S1D2 | $0.051 \pm 3.8 \times 10^{-4}$ | $\mathbf{0.034 \pm 8.5 \times 10^{-5}}$ |
| | S2D1 | $\mathbf{0.046 \pm 4.6 \times 10^{-4}}$ | $0.2 \pm 3.1 \times 10^{-2}$ |
| | S2D4 | $\mathbf{0.030 \pm 2.2 \times 10^{-4}}$ | $0.05 \pm 6.8 \times 10^{-3}$ |
| | S3D6 | $\mathbf{0.081 \pm 3.4 \times 10^{-4}}$ | $0.27 \pm 8.4 \times 10^{-2}$ |
| FEAT. SPAR. (%) | | $32.67 \pm 41.56$ | $\mathbf{52.21 \pm 6.37}$ |
| MODEL SPAR. (%) | | $67.05 \pm 21.31$ | $\mathbf{73.20 \pm 3.26}$ |
| SPUR. FEAT. SPAR. (%) | | $33.57 \pm 42.06$ | $\mathbf{52.39 \pm 6.36}$ |

Table 7: Effect of the regularization weight on performance and sparsity of different datasets.

| DATASET | L0 | C-NODE | PATHREG | LASSONET |
|---|---|---|---|---|

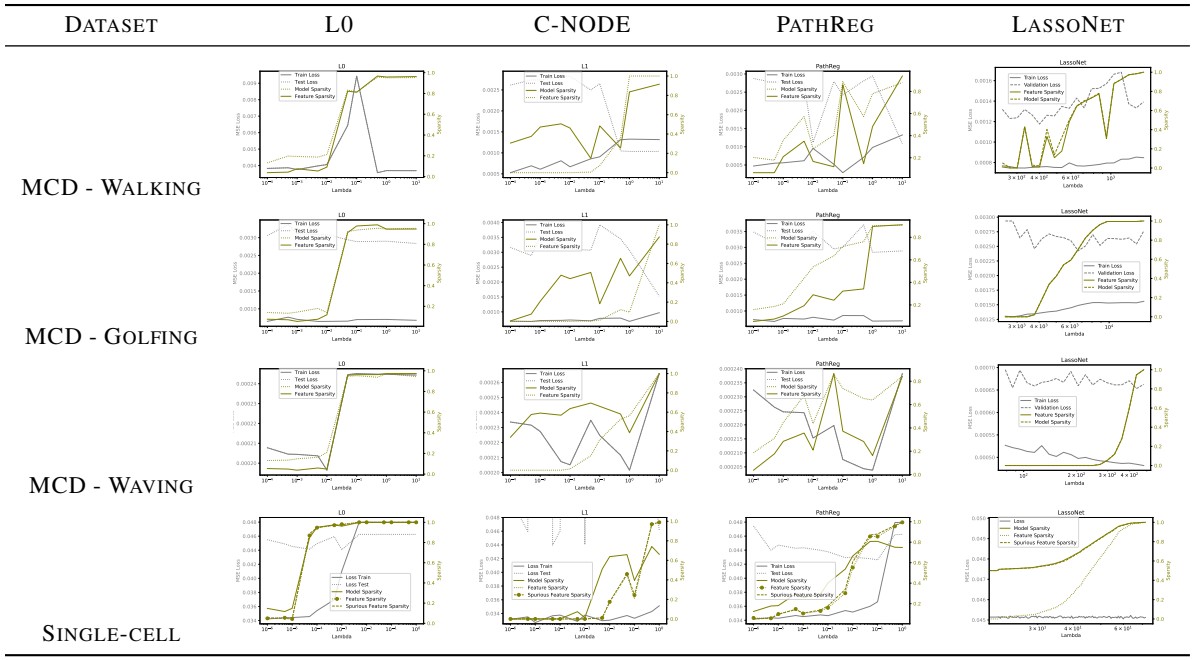