# OpenReview forum: "Sparsity in Continuous-Depth Neural Networks"
_NeurIPS.cc/2022/Conference — NeurIPS 2022 Accept_

### Official Review · Reviewer_Mra2 · 2022-06-18

**Rating:** 7
**Confidence:** 2
**Soundness:** 4 excellent
**Presentation:** 4 excellent
**Contribution:** 3 good

**Summary:**

The authors propose a method PathReg to improve sparsity in NODE models. The authors backup their claim that PathReg improve sparsity and performance by experiments in 3 different datasets.

**Questions:**

- The design of PathReg seems to be fairly straight forward with feed-forward networks. What is the reason you choose NODE-based models over general feed-forward networks?
- Gating are dependent on initialization so probably multiple experimental runs are needed for an accurate evaluation. How many times have you run each of your experiments?
- Gating could also potentially slow down the training process. What is the time consumption compared to other models?

**Limitations:**

As far as I know, there will hardly be any negative societal impact of the work.

**Strengths And Weaknesses:**

- The figures are nice and well-represent the experiments
- The effect of sparsification of PathReg is significant
- The performance of PathReg is descent.

---

> ### Author Response · Authors · 2022-08-02
> **Answers to Reviewer Mra2**
>
> We thank the reviewer for their time, excellent scores on Soundness and Presentation, as well as the constructive feedback.
>
> Regarding the 3 specific questions:
>
> **(1)** We agree that the usefulness of PathReg may not be limited to Neural ODEs as it can be applied also to, e.g., regular MLPs. In this work we focus on Neural ODEs, because we were mainly interested in the effectiveness of feature sparsity for dynamic systems identification as well as various types of (dynamic) out of distribution generalization. For these types of tasks, the sparsity we manage to enforce via PathReg is crucial. Regular feed-forward networks would not be able to capture time-series data like NODEs, but we agree that it would be an interesting direction for future work to analyze how PathReg may improve these arguably simpler models in tasks where OOD generalization or identification is important.
>
> **(2)** Indeed, initialization may matter. For this reason we ran each experiment at least 5 times. The variances across those runs with different initializations are shown in Tables 1-6 and we found them to be small enough to conclude that PathReg results are not notably affected by initialization.
>
> **(3)** Indeed, gating may result in slowdowns. Here are the timing comparisons to other models (not using gating), which we will also include in greater detail in the revised version: Execution time per epoch for the single-cell data: Base=0.9, L0 = 1.3, L1 = 0.95, PathReg = 1.05, GroupLasso = (not scalable), LassoNet = 0.83 seconds.

---

> > ### Comment · Reviewer_Mra2 · 2022-08-07
> > **Reply to Authors**
> >
> > Thanks for the clarifications! I think the work is promising, and I've increased my score.

---

> > > ### Author Response · Authors · 2022-08-09
> > > **Thank you!**
> > >
> > > Thank you very much for your time and increasing the score based on our rebuttal.

---

> ### Author Response · Authors · 2022-08-07
> **Thanks for reading our response**
>
> Dear reviewer,
>
> Thanks for reading and acknowledging our response.
>
> Best regards,
> the authors

---

### Official Review · Reviewer_ii69 · 2022-06-23

**Rating:** 6
**Confidence:** 3
**Soundness:** 3 good
**Presentation:** 3 good
**Contribution:** 2 fair

**Summary:**

This work proposes modifications (suggestions) to improve the performance of the Neural Ordinary Differential Equations towards better generalization. Specifically, for the neural network based function in X’=f(X, t), they propose a regularization technique for the weights of `f’ based on a combination of path-norm and L0 norm. The proposed regularization targets inducing feature sparsity, which ensures that the output variable depends on a few number of parents X_i’ = f(pa(X_i, t) and also the weight sparsity, which is traditionally shown to help with the generalization capability of the neural networks. This work primarily introduces a differentiable L0-based regularizer (PathReg) and extends it to NODEs. The authors argue empirically that the technique proposed by them improves over the state-of-the-art methods over a range of time-series tasks.

**Questions:**

Questions:
1. (Line 175) $W_j^(1)$ represents the first layer weights of $h_\theta$ ?
2. What are the choices of the distribution `q’ used for the L0 reg?
3. I understand that the equation 10 & 11 enforce the same constraint. In eq. (11), can you please expand on the form of the distribution $q^l( z | \phi)$ ? I believe that the $\phi$ params will be learned during the training process.
4. In equation (12), the expectation term samples `s’, as per my understanding, gives a set of G’s. Is the ||A_z|| term changing with every sample? How does the $\phi$ term come into picture here?

I request the authors to make these above points clearer in the paper as well. I went through the supplementary material too but I was not able to find clear descriptions of the procedure.

Experiment suggestions:
5. The \lambda_1 ||A||_1 term is introducing weight sparsity. How does the performance change if this term is dropped and only \lambda_0 ||A_z||_1 is used?


**Strengths And Weaknesses:**

(previous work summary) C-NODE uses the upper bound of L1(path-norm) and GroupLasso uses the group lasso norm as regularization. LassoNet introduces a skip connection and then enforces sparsity by putting constraints on the skip connection weights and the first layer of the feedforward NN.

Before I can assess the quality, novelty and impact of the work, I request the authors to address the points raised in the Questions & suggestions section. I am not able to follow certain aspects of the work.

Update: After the clarifications provided by the authors, I have updated my ratings.

---

> ### Author Response · Authors · 2022-08-02
> **Answers to Reviewer ii69--Part One**
>
> **Part One**
>
> We thank the reviewer for their time, constructive feedback, and especially their openness to further clarifications before making their decision! We’ll answer the concrete questions one by one.
>
> **(1)** Thanks for pointing that out. The notation across sections 2.2 and 3 indeed leaves room for ambiguity. We now explain the different $W$s involved in the different parts and will update our notation as well as exposition in the paper to remove ambiguity. At the risk of repeating points that were clear already:
>
> * In section 2.2, we introduce a single network $f_{\theta}$ mapping $n$ inputs $(X_1, \ldots, X_n)$ to $n$ outputs (the respective derivatives). The collection of all parameters of this MLP is $\theta$, which we split into the weights (and biases) for the individual layers denoted by $W^l$ for $l \in \{1, \ldots, L\}$ ($L$ is the number of layers).
>
> * In equation (3), we speak about $n$ different networks $f_{\theta_i}$ for $i \in \{1, \ldots, n\}$ mapping $n$ inputs to just one output each, i.e., one net for each variable. Here, the subindex of the weight matrix stands for the different networks. The expression $[W^1_i]_{\cdot, k}$ is the $k$-th column, of the first-layer weight matrix of the $i$-th network (i.e., the one predicting $\dot{X}_i$.
>
> * In the paragraph about LassoNet (starting at l.168), the overall function is now made up of two components: (1) a regular feed-forward network mapping $n$ inputs to $n$ outputs denoted by $h_{\theta}$, and (2) a separate $n\times n$ weight matrix $W_s$ corresponding to the “skip” weights (the subscript s has nothing to do with the subscript $i$ of $W$ in eq. (3)). The net $h_{\theta}$ maps X to the output non-linearly and $W_s$ can be thought of as mapping the $n$ inputs to the $n$ outputs linearly. We again partition the set of parameters $\theta$ of $h_{\theta}$ layer-wise into $W^{(l)}$ for layers $l \in \{1, \ldots, L\}$ (where now we inconsistently added parenthesis to the superscript, we will fix this). Note that $W_s$ is separate from those $W^{(l)}$.
> Now refers to the $j$-th column of $W^{(1)}$, again inconsistent with the notation we used in eq. (3) for the j-th column (will fix this).
> Also, it has nothing to do with indexing over multiple different networks. We will unify this. Hence $|W_j^{(1)}|\_{\infty}$ is the largest absolute weight “connected to the j-th input”. If this sup-norm is 0, the $j$-th input is entirely suppressed in $h_{\theta}$. Finally, there is a typo in the constraint of eq. (6), which should read (in the new notation, see below) $\|[W^{1}]\_{\cdot, j}\|\_{\infty} \le \rho \|[W_s]\_{\cdot, j}\|\_2$, i.e., the 2-vector-norm of the j-th column of the skip connection weights upper bounds (up to a scalar factor) the largest absolute possible weight in how the j-th feature enters the non-linear network $h_{\theta}$.
>
> ```
> To sum up, we will use the following unified notation in the final version (also highlighted in the revision):
> ```
> - Superscripts of $W$ will always be used without parenthesis and refer to layers in the network.
> -We will change the "skip weights matrix" $W_s$ to just $\mathcal{S}$ to avoid confusion with $W_i$, where the $i$ refers to different networks.
> - We will consistently use $[W]_{\cdot, j}$ for the $j$-th column of matrix $W$ to avoid confusion again with $W_i$, where $i$ refers to different networks.

---

> > ### Author Response · Authors · 2022-08-02
> > **Answers to Reviewer ii69--Part Two**
> >
> > **Part Two**
> >
> > **(2)** In this paper, we use binary concrete distribution (Maddison et al., 2016; Jang et al., 2016) as suggested by the original work (Louizos et al., 2018)[section 2.2].
> > A binary concrete random variable $s$ is distributed in the $(0, 1)$ interval with probability density $q(s|\phi)$ and cumulative density $Q(s|\phi)$. The parameters of the distribution are $\phi = (\log(\alpha), \beta)$, where $\log(\alpha)$ is the location and $\beta$ is the temperature.
> > It also performs as a smooth approximation to Bernoulli random variable allowing for gradient based optimization of its parameters through the reparametrization trick.
> > We will explain these in the appendix.
> >
> > **(3)** Thanks we will clarify this as well in the revision: Before eq. (11) we introduce gates for each layer (instead of just the input layer). The superscript $l$ in $q^l(z_{ij} \ne 0 | \phi_{i,j})$ corresponds to the layers $l \in \{1, \ldots, L\}$. Each $q^l$ in eq. (11) is then the same as the $q(z \ne 0 | \phi)$ as in eq. (5), i.e., determined again by the CDF of $q(s | \phi)$ being a concrete distribution.
> >
> > **(4)** This is true. Sampling $s$ from $q(s|\phi)$ can result in different $G$ and $A_z$ respectively. The recent work “Winning the Lottery with Continuous Sparsification” by Savarese et al. (2020) suggests a method that does not require sampling. While we believe this is an interesting follow-up research, we have not observed the “sample dependence of the regularization terms” to hinder performance or lead to inconclusive results over multiple runs. Typically, we find $q(s|\phi)$ to saturate during training and thus little variation across $A_z$ during later stages of the training. Moreover, we can infer the uncertainty of causal graphs using the parameters of the distributions.
> >
> > **(5)** So far, we observe higher overall sparsity by regularizing both the weights and the gates: For the single-cell experiment, the best performing model using both regularizers has MSE=0.0351 and feature sparsity=88.36\% (Table 5). Using only $||A_z||_1$, the best model has MSE=0.0356 and feature sparsity=79.14\%. Thanks for the suggestion. We will add the results only using  $\lambda_0 ||A_z||_1$ to the paper in greater detail as the reviewer suggests.
> >
> > To summarize, we will fix all ambiguities in terms of superscripts with and without parenthesis for layers, different meanings of subscripts of weight matrices as columns/different networks/just a label as in $W_s$. Finally, we’ll add more details from the original LassoNet and L0 (distribution choices, etc.) papers to make our work more self-contained.

---

> > > ### Comment · Reviewer_ii69 · 2022-08-07
> > > **[Response to authors]**
> > >
> > > I appreciate the authors clarifying the notations and answering my questions. After due considerations, I feel that the contribution of this work is incremental over the previous related methods. I will update my ratings according to the updated draft.

---

> > > > ### Author Response · Authors · 2022-08-09
> > > > **Thank you!**
> > > >
> > > > Thank you very much for your time and consideration!

---

> ### Author Response · Authors · 2022-08-07
> **Curious about your assessment**
>
> Dear reviewer,
>
> We would like to thank you once again for your time, valuable feedback, and patience in awaiting our clarifications.
> We tried hard to answer all your questions and believe they already helped us to substantially improve our paper.
> As the author-reviewer discussion period approaches, we are now very curious to hear about your assessment.
>
> Best regards,
> the authors

---

### Official Review · Reviewer_hJba · 2022-07-12

**Rating:** 7
**Confidence:** 3
**Soundness:** 4 excellent
**Presentation:** 4 excellent
**Contribution:** 3 good

**Summary:**

In this work, the authors propose a novel feature regularization method called PathReg for NODEs, a neural net architecture that can reconstruct and extrapolate time series data. This method allows the NODE to learn parameters with 2 types of sparsity: 1) weight sparsity and 2) feature sparsity. While methods exist to regularize NODEs with feature sparsity (C-NODE and Group Lasso), there is currently no approach that also integrates weight sparsity. The primary contribution of this work is to first adapt two methods for weight sparsity in regular neural networks to NODEs (L0 and LassoNet) and then combine both types of sparsity in PathReg. The authors evaluate their method on a set of convincing simulated and real datasets against several baselines using reasonable metrics. Furthermore, they provide some interesting insight on the data by interpreting the feature correlations derived from their model. The results and experiments described in the manuscript are quite promising. However, there are several claims that need to be clarified and additional experiments that would be needed to support claims or "nice to have" to improve the paper. Details are given below.

**Questions:**

See above.

**Limitations:**

Yes

**Strengths And Weaknesses:**

(1)
First, we very much appreciate the author’s thorough evaluation of the existing literature in this space and the clear statement of their contribution. It was easy to recognize exactly how the author’s are uniquely innovating in the context of other approaches in the field.
(2)
The author’s describe several advantages of sparsity. These include: 1) improved speed and efficiency, 2) better generalization, and 3) improved interpretability.
(a)
Would it be possible to show runtime benchmarks for each method? The reason for requesting this benchmark is that C-Node performs comparably (and sometimes better on certain datasets) on the reconstruction metrics. One reason to justify the use of PathReg would be the runtime. Is it the case that learning fewer weight parameters by combining both feature and weight regularization makes the approach faster?
(b)
Is it possible to perform a similar interpretation on the A matrix in C-NODE as was done on the A matrix in PathReg? If not, can the interpretability of C-NODE and PathReg results be compared in some other way?
(3)
It is not completely clear what the overarching premise of combining both feature and weight sparsity actually achieves in practice. In other words, why should one care that the weight metric is higher for PathReg in cases when the generalization error is comparable.
(4)
It is not clear what Method Ranking and Overall Score mean in Figure 3 and 6. Regardless, these are not useful metrics and could be discarded to make the figures more clear.
(5)
It is not clear what path weight means (x-axis in right most plot in Figure 2).
(6)
The interpretation of the correlation matrix A for both the motion capture data and single cell data are quite exciting. We would encourage the authors to consider adding some examples to the main text and to elaborate on subjective claims such as “learn sensible and arguably correct adjacency matrices (derived from joint connectedness)”. For example, were leg and arm joints more connected in the walking data (where both appendages move) compared to the golf data (where legs do not move). In general, we did not find Figure 4 or Figure 5 (top) to be particularly useful for illustrating the main claims in the paper and could be replaced.
(7)
The C. elegan embryo development dataset in Packer et al. 2019 has single cell time series data with experimentally derived time. This could be used instead of the inferred pseudotime. The approach used to infer pseudotime is also not currently state of the art. Please refer to Saelen’s et al. 2019 for thorough benchmarks of over 70 methods.
(8)
RNASeq batches should be corrected for batch effect. Consider methods like COMBAT or RUVSeq.
(9)
It is not clear how spurious features are defined in Figure 6. How did the authors determine that a pair of genes did not directly affect each other’s expression? What does it mean for a transcription factor to be inferred by PathReg as “important”? One approach to determine whether certain TFs affect other genes is to take knockout data (such as from ENCODE) to perform differential expression between control and TF KO data.
(10)
Please elaborate on the contribution for releasing data. What aspect of the data was modified to warrant re-release under this publication?
(11)
The Base model has significantly worse generalization error compared to other methods for gene expression. This was not so much the case with the motion data where the generalization error was comparable to the other methods for some datasets. This is quite an interesting result. Why do the authors think this is the case and can their model’s improved interpretability aid in figuring out the answer?

Minor errors and comments:
-----------------------------------------
In the Figure 1 feature sparsity figure, shouldn’t x0 in the input be connected to x0 in the output? How can it be possible for a feature to have no effect on reconstructing itself? Even though this is a cartoon, this is a bit misleading.

Typo line 241: PsthReg -> PathReg

Perhaps highlight that S1D2 was the training data in Figure 6

Is the feature sparsity % of exactly 75.52 duplicated multiple times in Table 1 correct?

---

> ### Author Response · Authors · 2022-08-02
> **Answers to Reviewer hJba--Part One**
>
> **Part One**
>
> We thank the reviewer for their time and the extremely thorough review with lots of valuable and constructive feedback. We’ll address the questions and comments one by one:
>
> **(1)** Thank you.
>
> **(2)** **a.** Unfortunately, the *training* speed of PathReg (1.05sec per epoch for the single-cell data) is slightly worse than that of C-NODE (0.95sec per epoch) because of the learned L0 gates, which add computational overhead (we will add timings for all methods in the revision). However, C-NODE has several other limitations, which we overcome with PathReg. A key criticism of C-NODE is that the weights do not typically converge to exact zeros and a manually chosen cut-off is required to determine which are interpreted as zero (no causal parents) and which aren’t (causally relevant parents). Crucially, even when we interpret a small weight to be “essentially zero” the corresponding input still affects all outputs (the weight is never exactly zero in the forward pass). PathReg on the other hand encourages gates to become exactly zero in the sense that with overwhelming probability we propagate exact zeros as input. Moreover, we may use the probability distributions of gates in PathReg to evaluate the graph uncertainty in a more meaningful way than, say, trying to attach meaning to weights smaller than an arbitrary absolute threshold value in C-NODE. We will include quantitative runtime benchmarks in the revision.\
> **b.** The adjacency matrix A in Eq. 2 of C-NODE is inferred directly from the weight matrices. An “input-output” connection is said to exist, if the corresponding entry in A is non-zero (or above some threshold). PathReg regularizes $A_z$ instead, where the ij-th entry of $A_z$ corresponds to the “probability that some non-zero paths from input j to output i are allowed in the network”. While not exactly equal, we believe that we can interpret A in C-NODE and A_z PathReg similarly, but PathReg even yields a more meaningful quantification: An exact zero at position ij in A or $A_z$ means that j does not influence i in both methods. A very small value epsilon in $A_z$ means that j does not influence i with probability 1 - epsilon, whereas an entry epsilon in A (of C-NODE) would have to be interpreted relative to the other effect sizes, etc.
>
> **(3)** From a practical perspective, we believe the interesting observation is that we can obtain the complementary benefits of weight sparsity (can use this for pruning downstream, i.e., memory reduction) as well as feature sparsity (interpretability) while also achieving truly synergistic effects in terms of generalization/identifiability beyond what each method alone can do. Here, the “typical colloquial effect” of both types of regularization also make sense (weight sparsity: “reducing overfitting” -> generalization; feature sparsity: “capturing only most relevant/strong interactions” -> generalization).
>
> **(4)** The overall score is the average over all the scores for each method across datasets and we then ranked the methods accordingly. Our goal here was mostly to provide the reader with some “main takeaways”, but we agree that they do not strictly add much to the more fine-grained results.
>
> **(5)** For “paths” in Figure 2 (right), the histogram is over the entries of the matrix in Eq. (13) in Section 3.3. This matrix essentially contains all “aggregated (absolute) path weights” from a given input to a given output already taking into account the gates. Note that all entries of this matrix (on the x-axis) are nonnegative and there are 529x529 of them. We will include a more detailed description in the Figure caption.
>
> **(6)** Thanks a lot for all these suggestions and insights. Overall, we tried to refrain as much as possible from “visual claims of correctness” (especially when ground truth is not known; we will drop the remaining subjective claims like the one mentioned), and we are happy to discuss the content of these figures in more quantitative ways in the revision. We’ll also bring one (or multiple, as space permits) of the mentioned figures that allow for a sensible interpretation to the main body of the paper in replacement for Fig 4 and/or 5. We also found those “connectivity” results exciting!

---

> > ### Author Response · Authors · 2022-08-02
> > **Answers to Reviewer hJba--Part Two**
> >
> > **Part Two**
> >
> > **(7)** One of the main issues we had with actual time-series datasets is that often the time points are too coarse grained and therefore, we may still need to perform some type of pseudo ordering to infer the order of cells within the observed time steps. We still think this is a great suggestion and will continue working on this to apply the method in “actual time”. We have indeed compared various pseudo time methods during our experimentation phase.From our results, it appears that diffusion pseudo time (DPT) performs equally well  (for this dataset and our purposes) as newer methods such as scvelo. Moreover, the inferred lineages and the order of cell types make sense biologically, which is why we stuck to DPT for the main paper. We are happy to include comparative results using other pseudo time methods upon request.
> >
> > **(8)** Good point. While we expect batch correction to “help” our method in that it should be easier to learn dynamics overall, we intentionally avoided batch correction, interpreting each batch as some form of OOD experiment to study the generalization of the methods. However, we agree that it will be interesting to compare what happens with and without batch correction. We have added this to the list of follow-up experiments we will try to get ready in time for a revision.
> >
> > **(9)** We used the data-driven lists of transcription factors and target genes inferred using CellOracle and paired ATAC-seq data of the same samples (which we briefly explained in the appendix). We then defined the interactions between target genes not annotated as transcription factors (TF) as spurious, given their lack of biological support for direct gene regulation. Our main intuition is that a TF is important for a model if it affects at least a gene g, or more specifically the TF enters the component of $f$ corresponding to gene g. Knocking out important genes to validate the models is also a very interesting direction, however, we don’t have this readily available for the dataset shown in the paper and will have to leave this to future work.
> > Clearly, we have no guarantees of our choice of “spurious” being “correct” and it certainly does not capture all types of spurious associations. We will add some more details and will properly define what we mean by spurious in this case in the revision together with a word of caution.
> >
> > **(10)** Our goal here is to release the combination of datasets from different domains in a pre-processed format that can be readily picked up easily by anyone working on time-series modeling, system identification, or causal inference in dynamical systems without having to think too much about appropriate pre-processing in the respective domains. This comes from a conviction that especially in the latter field we need to start to move away from purely simulated data and embrace some real-world data instead (even if interpretation and verification is difficult).
> >
> > **(11)** We found it difficult to make conclusive claims here. Our working hypothesis is that both the base model and C-NODE perform worse on the single-cell data, because (a) the dataset is much larger than the other two examples (in terms of variables), and (b) we are training the model simultaneously for three lineages. Both (a) and (b) may “lure” existing techniques into overfitting (relatively few timesteps provided to infer the relationships between many variables) and that they have trouble effectively pruning connections. Claiming that the interpretability of our method (while valuable in and of itself) can also shed light on the underperformance of the other methods at this point would be pure speculation on our end.

---

> > > ### Comment · Reviewer_hJba · 2022-08-09
> > > **Reply to author's rebuttal**
> > >
> > > We appreciate the detailed response. It would still be good to see the authors include in the revision specific suggestions they made above which are still not available in the current updated version (points #6 #8 above).

---

> > > > ### Author Response · Authors · 2022-08-09
> > > > **Finishing touches**
> > > >
> > > > Thanks for picking this up again.
> > > >
> > > > ### Regarding 6
> > > >
> > > > Besides running the other experiments (more below), we also worked on interpretations of the motion capture data along the lines you suggested. Beyond the detailed results (Figure 8) and interpretations thereof (Figure 8 caption) for the motion capture data in the supplement, we will further polish and extend that Figure (add further annotations and types of movement) for the final version and add more explanations, e.g.: "For example, when waving with the left hand, PathReg correctly identifies only interactions between joint points corresponding to the left arm. For the golfing moment, PathReg correctly recovers that both arms move in a connected fashion, but legs and most other body parts are not substantially involved in the movement." (See below about adding these to the main.)
> > > >
> > > > Regarding Figures 4 and 5, we are a bit torn in that we believe that just "visualizing the data" may be helpful, particularly for readers without experience in single-cell and/or motion capture data. At the same time, we agree that they (especially Figure 4) take up valuable real-estate without communicating noteworthy findings or results by themselves.
> > > >
> > > > Since an additional content page will be allowed for accepted papers but apparently not the revisions during the discussion phase, we are planning to reduce and compress Figure 4 (and maybe 5) to take up less space (but still keep them) and move the interpretations of the correctness of our findings (including a condensed version of Figure 8 in the appendix and potentially more such findings) from the supplement to the main text.
> > > >
> > > > ### Regarding 8
> > > >
> > > > We have been actively working on this experiment. Since the batch corrected data that we used is not publicly available and the batches are annotated individually, we have thus far been using the ScanVI tool to remove the batches. In our results up until now we don't observe major batch effects, in particular the expression of individual genes along lineages, which we use as input to our model, appear to be fairly robust. However, the full validation including publication-ready figures will take us some more time and we are confident that we can present the results in the final revision.

---

> > > > ### Author Response · Authors · 2022-08-09
> > > > **Updated revision**
> > > >
> > > > We just uploaded another revision of our manuscript. In particular, we added drafts of how we will address
> > > >
> > > > * (2) added runtimes in Appendix A
> > > > * (4) caption Figure 3
> > > > * (5) caption Figure 2
> > > > * (6) Appendix C.2
> > > > * (9) Appendix B.3
> > > >
> > > > but will add more details and polish the writing for the final revision. We had to add some of the improvements to the appendix for now, as the page limit during the author/reviewer discussion period is still 9 pages. Together with the improvements (and space reductions) discussed in our previous comments (on 6) and 8)), we will try to fit as many of these updates in the main text of the camera-ready version (which allows for 10 pages) should the manuscript get accepted.
> > > >
> > > > Thanks again for all the useful comments and for helping us to substantially improve our manuscript.

---

### Official Review · Reviewer_388f · 2022-07-19

**Rating:** 5
**Confidence:** 3
**Soundness:** 3 good
**Presentation:** 3 good
**Contribution:** 1 poor

**Summary:**

This paper focuses on identifying and empirically verifying the ways that Neural ODEs can be regularized towards improved generalization. Sparsity in weights and features are explored in particular, where a new regularization technique called PathReg is developed as an analogue to other common regularization techniques in deep neural network training. There are experiments performed on mocap and RNA-sequencing datasets to assess the generalization capabilities of continuous-depth neural networks.


**Questions:**

Do these changes lead to more favorable ODE solvers over others? What are the speedups in general?

Would you be able to compare again other neural ODE techniques and summarize their differences.


**Limitations:**

Not discussed.

**Strengths And Weaknesses:**

## Novelty, relevance, significance
The work is a thorough empirical survey of the proposed PathReg method which is a unification of L0 regularization (which specifying the weights in a neural net) and LassoNet (which selects features based on how much an input affects the output) with residual weight regularization. The loss is modified in the usual way to include these extra regularization terms and generally the results point at the fact that combining feature and model selection, building upon improvements shown by either methods already achieved independently.

I generally find data augmentation types of papers difficult to rate in terms of high novelty. It also makes sense that better out-of-distribution generalization is achieved when the model is not trained to over-fit to in distribution and hence can better ignore spurious features.

## Soundness / Correctness & Clarity
The derivations for the new loss and adaptation of existing two methods for regularization / sparsification are generalized correctly to the NODEs setting. L0 regularization of terms is also standard and hence sound.

## Quality of writing/presentation
Writing and presentation are well-done. Figures are clear and labeled.

---

> ### Author Response · Authors · 2022-08-02
> **Answers to Reviewer 388f**
>
> We thank the reviewer for their time and feedback as well as their positive judgment of the soundness and presentation of the paper. While the high level connection between “avoiding overfitting” and “better generalization” is generally accepted, that does not imply that simply any regularizer will effectively address overfitting and thus improve generalization. Moreover, for interpretability and OOD generalization the connections to regularization and how to regularize effectively are still underexplored.
>
> The main contribution of our paper is a simple, effective, and scalable feature sparsity regularizer that does outperform state-of-the art techniques. In particular, some of the other possible regularization schemes we compare to do not lead to a comparably good OOD generalization.
>
> Indeed, there are only few existing feature sparsity methods and they are often not scalable to large datasets with hundreds to thousands of variables. Through our thorough evaluation, we show that PathReg can easily scale up without losing performance.
>
> We did not fully understand the concern about how to rate “data augmentation types of papers” in terms of novelty, since we do not see how our paper is a “data augmentation type of paper”. We neither propose a data augmentation method, nor are we using data augmentation in our work.
>
> Regarding the two questions:
>
> **(1)** Speed-up and favorable ODE solver:\
> It has been shown in the related work (which we also discuss in the introduction) that regularization can improve the number of model evaluations for a single step of the ODE solver. While this is important, our main objectives are improving identifiability and dynamical system inference as well as generalization. Hence, work on speeding up NODEs (either by direct improvements of ODE solver or by reducing the number of function evaluations during training), is complementary to our findings and our method will benefit from such improvements. Still, we show that PathReg is easily applicable to large-scale datasets, for example single-cell RNA-seq with hundreds of features, even without extensive work on improving ODE solver efficiency. The summary of execution times per epoch for the single-cell data running on a regular CPU (which we will include in greater detail in the revision) are as follows (all in seconds): Base = 0.9, L0 = 1.3, C-NODE = 0.95, PathReg = 1.05, GroupLasso = (not scalable), LassoNet = 0.83 seconds. While LassoNet is the fastest, its convergence speed is noticeably lower than the rest.
>
> **(2)** Comparison with other neural ODE techniques: \
> In our paper, we compare PathReg to 5 competing methods including Vanilla neural ODE, C-NODE, and GroupLasso for neural ODEs. We explained the differences to other neural ODE regularizers (specifically the ones aiming for “easy to solve” neural ODEs, i.e., few function evaluations, which are orthogonal to our work) both in the introduction and Section 2.2. We will expand on those differences in the revision to improve the clarity of our paper.

---

> > ### Comment · Reviewer_388f · 2022-08-09
> > **response to authors**
> >
> > Thanks for answering my questions in detail, I appreciate the summary of the differences with other forms of ODE regularization and acknowledge that this is an orthogonal approach to regularization in "easy to solve" works. I have increased my score, though I still feel that this work is incremental since many works have commented on the effectiveness of optimizing simpler NODEs, whether that is in terms of simpler dynamics to solve over or additional regularization terms. Prior works such as LassoNet show the effectiveness of incorporating sparsity during training and it is not particularly surprising that the application of such here begets the same outcome.

---

> > > ### Author Response · Authors · 2022-08-09
> > > **Thanks for your engagement!**
> > >
> > > Thank you very much, we appreciate that reviewers are taking time from a possibly very busy schedule to engage in numerous discussions at NeurIPS and show willingness to update their scores based on the rebuttal!
> > >
> > > In further response to your comments, we would like to emphasize again that our proposed method (PathReg) is not akin to LassoNet in how it works (even though you are right of course in that they both broadly aim at enforcing sparsity).  In particular it is not merely a combination of existing regularizers. Also, it is not clear that any type of sparsity regularization will help. For example, the LassoNet extension to NODEs does not perform well in our dynamical systems (see summary result figures and tables). With PathReg we have developed not just "yet another" sparsity regularization for NODEs, but one that also works well and introduces new ideas (regularizing "paths through neural networks"). Again, thanks for your feedback and engagement!

---

> ### Author Response · Authors · 2022-08-07
> **Awaiting further feedback**
>
> Dear reviewer,
>
> We would like to thank you once again for your time and feedback.
> As we believe we made several important points worth discussing or acknowledging and had a question about your comments, we would like to kindly request further feedback as the deadline for the end of the author-reviewer discussion period approaches.
>
> Best regards,
> the authors

---

### Meta-Review · Area_Chair_gMD3 · 2022-08-26

**Recommendation:** Accept
**Confidence:** Certain

**Metareview:**

The paper proposes a new sparsity inducing regularization scheme for continuous-depth neural networks. The reviewers acknowledged the relevance of the proposed method and generally appreciated the results. The paper is nicely written and provides a range of interesting experiments demonstrating the effectiveness of the proposed method. I want to thank the authors for their detailed responses that helped in answering some of the reviewers' questions. (The reviewers have provided detailed feedback in their reviews, and we strongly encourage the authors to incorporate this feedback when preparing a revised version of the paper. I also would like to encourage the authors to carefully revise the related work section on  continuous-depth neural nets to better acknowledge work that has appeared in the last 2 years.) In summary, the feedback of the reviewers is positive and thus I recommend accepting this paper.

**Award:**

No

---

### Decision · Program_Chairs · 2022-09-14

Accept